# Comprehensive Evaluation of Hepatotoxicity Following Radiation Therapy in Breast Cancer Patients

**DOI:** 10.3390/cancers17193252

**Published:** 2025-10-08

**Authors:** Jun Yeong Song, Soon Woo Hong, Sang-Won Kang, Bum-Sup Jang, In Ah Kim

**Affiliations:** 1Department of Radiation Oncology, Seoul National University College of Medicine, Seoul 03082, Republic of Korea; james94@snu.ac.kr (J.Y.S.);; 2Department of Radiation Oncology, Seoul National University Bundang Hospital, Seongnam 13620, Republic of Korea; 3Department of Radiation Oncology, Seoul National University Hospital, Seoul 03080, Republic of Korea

**Keywords:** breast cancer, radiotherapy, liver enzyme elevation, radiation-induced liver disease, hepatotoxicity

## Abstract

Breast cancer treatments often include radiation and chemotherapy that may affect the liver. Because the liver is close to the breast and processes many drugs, there has been concern about possible damage. In our study of over 500 patients, only a small number showed mild and temporary changes in liver function tests, and very few developed serious liver problems. These results suggest that modern breast cancer treatments are generally safe for the liver. Still, patients who receive chemotherapy before surgery may need closer monitoring to keep their liver healthy during treatment.

## 1. Introduction

Breast cancer was the second most common cancer worldwide in 2022, with more than 2 million new cases [1]. Because of this prevalence, it is estimated that more funds, manpower, and time are invested to fight breast cancer than any other type of cancer [2]. Consequently, numerous novel treatment approaches across various modalities such as radiation therapy (RT), chemotherapy, endocrine therapy, targeted therapy, and immunotherapy have emerged. However, the current clinical trial designs evaluating systemic therapies rarely consider the integration or timing of RT, and guidelines such as the ACROP-ESTRO consensus highlight the lack of evidence on the safety of combined modalities [3].

Due to advances in treatment and early detection, breast cancer has a favorable prognosis, with a 5-year overall survival rate of 93.8% in South Korea [4]. As survival improves, minimizing treatment-related adverse effects has become increasingly important. To minimize adverse effects, treatment decisions should consider patient-specific factors such as age, comorbidities, and cancer stage.

Liver toxicity is among the adverse effects to consider. Because of its role in metabolizing drugs and its proximity to the breasts, the organ is susceptible to toxicities from both local and systemic treatments for breast cancer. Furthermore, deteriorating liver function can delay further cancer treatment and cause complications in multiple organs. Radiation-induced liver disease (RILD) can occur 2 to 3 months after radiation therapy (RT). Classic RILD presents with elevated levels of alkaline phosphatase (ALP) and is pathologically characterized as hepatic veno-occlusive disease (VOD) [5]. On the other hand, patients who develop non-classic RILD have underlying chronic liver disease and show remarkably elevated serum levels of transaminases. Previous studies suggest that radiation dose and spared liver volume are associated with the occurrence of RILD [6,7,8,9]. Additionally, systemic therapies used against breast cancer often cause hepatic toxicity. Chemotherapy, endocrine therapy, and targeted therapy can cause hepatotoxicities such as elevated liver function test (LFT) scores, VOD, and hepatitis [10].

We investigated the impacts of multimodal treatments on liver toxicities in patients with breast cancer to determine the extent to which various treatments impact the liver. Also, by determining which modalities or regimens are associated with the incidence of liver toxicity, we aimed to suggest precautions to follow when treating patients with breast cancer.

## 2. Materials and Methods

This study was designed as a multicenter retrospective study. This study was approved by the Institutional Review Board of each participating institution. Patients satisfying the following criteria were included: (1) received surgical resection for breast cancer; (2) received postoperative RT in 2021; (3) performed LFT prior to RT, after other treatments were concluded; (4) performed LFT 5 to 7 months after RT; (5) no history of prior abdominal or pelvic RT; (6) no history of liver disease or viral hepatitis; (7) underwent planning CT that included the entire liver volume. Also, body habitus and breast size were not used as selection criteria.

Details of systemic therapies administered and clinical factors such as age, sex, tumor size and stage, and laterality of the breast cancer were collected. Systemic therapies included regimens of cytotoxic chemotherapy, endocrine therapy, and targeted therapy. Details of RT, such as dose, fraction, boost dose, boost fraction, and RT technique (e.g., 3D-conformal radiation therapy [3D-CRT] and intensity modulated radiation therapy [IMRT]) were also collected. RT were planned and delivered under free-breathing conditions without the use of deep inspiration breath hold (DIBH) or other respiratory motion management techniques. RT target volumes were delineated according to the consensus guideline provided by the Radiation Therapy Oncology Group (RTOG) Breast Cancer Atlas for Radiation Therapy Planning [11]. From the RT profiles, dose-volume histogram (DVH) parameters such as liver volume receiving 5 Gy or greater (V5Gy), V10Gy, V20Gy, and mean liver dose were collected for each patient. To enhance the understanding of the RT administered in this study, an image from the simulation CT of a representative case is presented in Figure 1, with the 5 Gy, 10 Gy and 20 Gy isodose lines delineated.

From the LFTs performed before and after RT, results for aspartate aminotransferase (AST), alanine aminotransferase (ALT), and ALP were collected. The upper limits of normal values for AST, ALT, and ALP were 40 U/L, 40 U/L, and 115 U/L, respectively. Liver toxicity was evaluated according to the Common Terminology Criteria for Adverse Events (CTCAE) grade (version 5.0) for the liver enzymes. Liver enzyme elevation (LEE) was defined as a change in the CTCAE grade of AST, ALT, or ALP. ‘Any LEE’ was defined based on the maximum change in CTCAE grade among AST, ALT, or ALP. The occurrence of LEE before and after RT was assessed. Classic RILD was diagnosed when ALP levels were at or above twice the upper normal limit. Non-classic RILD was diagnosed when AST or ALT levels were at or above five times the upper normal limit.

Pearson’s chi-square test or Fisher’s exact test were used to compare differences in categorical variables. The optimal cutoff of DVH parameters that maximizes the area under the curve of the receiver operative characteristic curve was estimated using the Liu method to convert continuous variables to categorical variables [12]. Logistic regression was used in univariate and multivariate analyses to investigate the associations between LEE and various clinical factors and DVH parameters. *p* values and 95% confidence intervals (CIs) were calculated, and results were considered significant if the *p* value was < 0.05. All statistical analyses were performed using the Stata software (version 17; StataCorp LLC, College Station, TX, USA).

## 3. Results

### 3.1. Baseline Characteristics

Five hundred twenty-nine patients with breast cancer met the eligibility criteria and were included in the analysis. The clinical factors and treatment profiles of patients are shown in Table 1. The patients’ age ranged from 27 to 86 years, with a median of 51 years. Pathologic stages T1 and N0 accounted for 317 (59.9%) and 315 (59.5%) patients, respectively. Three hundred ninety-two (74.1%) patients had right-sided breast cancer, and 137 (25.9%) patients had left-sided breast cancer. All the patients received surgical resection. Four hundred fourteen (78.3%) patients received breast-conserving surgery, and 353 (66.7%) received selective lymph node biopsy. Two hundred sixty-one (49.3%) patients received cytotoxic chemotherapy, 359 (67.9%) received endocrine therapy, and 79 (14.9%) received targeted therapy. Among the cytotoxic chemotherapies, taxane, cyclophosphamide, Adriamycin, and platinum-based agents were given to 257 (48.6%), 240 (45.4%), 135 (25.5%), and 43 (8.1%) patients, respectively. Prescribed radiation dose ranged from 38.5 Gy to 50.06 Gy, and more than half of the patients were prescribed 42.56 Gy. Majority of the patients (83.7%) received boost RT. The median values of the mean dose to the liver for V5Gy, V10Gy, and V20Gy were 168.9 cGy, 6.1%, 0.75%, and 0.04%, respectively. Four hundred fourteen (78.3%) patients received IMRT, and the rest received 3D-CRT.

### 3.2. LEE and RILD

LEE of AST, ALT, and ALP before and after RT was analyzed, and the results are shown in Table 2. Grade 1 LEE of AST was observed in six (1.1%) and eight (1.5%) patients before and after RT, respectively (*p* < 0.001). Grade 1 LEE of ALT was observed in 15 (2.8%) and 10 (1.9%) patients before and after RT, respectively, and grade 2 LEE of ALT was observed in 2 (0.4%) patients after RT (*p* < 0.001). Grade 1 LEE of ALP was seen in 23 (4.3%) and 32 (6.0%) patients before and after RT, respectively (*p* < 0.001). Liver enzymes were elevated to a degree that met the diagnostic criteria of RILD in three (0.6%) patients, one of which displayed classic RILD, whereas the other two displayed non-classic RILD.

Two patients were diagnosed with non-classic RILD. Upon diagnosis, systemic therapy was discontinued in both cases. One patient was treated with ursodeoxycholic acid (UDCA) and silymarin, while the other received UDCA in combination with Godex^®^ capsules. Ultrasonography was performed in one patient ten days after the diagnosis, which revealed diffuse fatty liver. One patient who was diagnosed with classic RILD was admitted for generalized edema with seizure-like movement. But no specific treatment was given for RILD. It resolved concurrently with the improvement of the underlying diseases, and no imaging was done for RILD.

Results of the analysis of changes in the CTCAE grade of each liver enzyme are shown in Table 2. Six (1.1%), 9 (1.7%), and 25 (4.7%) patients showed CTCAE grade increases in AST, ALT, and ALP, respectively, with all but 1 patient showing grade 1 increases.

### 3.3. Univariate Analysis of DVH Parameters

Results of the analysis of associations between LEE of each liver enzyme and various DVH parameters are shown in Table 3. There were no significant associations between changes in DVH parameters and any LEE or LEE of AST, ALT, or ALP. Although the median values of DVH parameters were larger when LEE occurred, changes in DVH parameters did not translate into a significant difference in any LEE (mean dose, *p* = 0.196; V5Gy, *p* = 0.218, V10Gy, *p* = 0.336; V20Gy, *p* = 0.996).

To analyze the DVH parameters as categorical variables, we estimated optimal cutoffs using the Liu method. The optimal cutoffs for mean dose, V5Gy, V10Gy, and V20Gy were 135.2 cGy, 3.54%, 1.74%, and 0.01%, respectively (Appendix A). The associations between LEE of each liver enzyme and the DVH parameter values categorized by the optimal cutoffs were analyzed, and the results are shown in Table 4. When LEE of AST, ALT, or ALP were analyzed separately, there were no significant associations with DVH parameters; however, any LEE was significantly associated with mean dose and V5Gy but not with V10Gy and V20Gy (mean dose, *p* = 0.040; V5Gy, *p* = 0.036; V10Gy, *p* = 0.156; V20Gy, *p* = 0.166).

We have selected four representative cases with high mean liver dose and V5Gy, contoured and recalculated the DVH according to the ESTRO guideline, and compared it with the pre-existing RTOG guideline-based DVH parameters (Appendix A). Three out of four cases demonstrated a reduction in V5Gy; however, the changes in V10 Gy and V20 Gy were not consistent across all 4 patients (Appendix A).

### 3.4. Multivariate Analyses

Multivariate analyses were performed to investigate associations between any LEE and DVH parameters, RT technique, the use of neoadjuvant therapy, and various clinical factors (Table 5). Each of the DVH parameters and clinical variables were analyzed separately. As a result, the use of neoadjuvant therapy was significantly associated with any LEE in analyses with all the DVH parameters. However, there were no significant associations between any LEE and other variables such as laterality, DVH parameters, RT technique, and the use of cyclophosphamide, taxane, platinum, or tamoxifen.

## 4. Discussion

We investigated the profiles of hepatotoxicity in 529 patients with breast cancer and their associations with the treatments received. According to the CTCAE grade, 6 (1.1%), 9 (1.7%), and 25 (4.7%) patients experienced at least grade 1 elevation in AST, ALT, and ALP, respectively. RILD was observed in three (0.6%) of the patients. Any LEE was significantly associated with mean dose and V5Gy but not with V10Gy and V20Gy. In the multivariate analysis, DVH parameters and RT technique were not associated with the occurrence of any LEE.

Several studies have demonstrated that incidental liver irradiation during RT for breast cancer may result in subclinical LEE, with mean liver dose and V5 identified as contributing dosimetric factors. A multicenter study involving 100 right-sided breast cancer patients reported approximately 15% increases in AST, ALT, and gamma-glutamyl transferase following RT, and found that a mean liver dose exceeding 2 Gy was associated with these changes [13]. Similarly, Lauffer et al. observed correlations between low-dose liver exposure and alterations in hepatic enzymes during right-sided breast RT [14]. These findings are consistent with the present study, in which both mean liver dose and V5 were significantly associated with any LEE in univariate analysis. However, in the multivariate analysis of our study, neoadjuvant therapy remained a sole independent predictor of LEE, suggesting that the impact of liver dose may be confounded by systemic therapy. Notably, all studies reported comparable liver dose ranges and observed only low-grade, transient LEE without RILD. Collectively, these results imply that while DVH parameters may contribute to LEE, their clinical significance is limited, particularly in the context of modern multimodal treatment.

In this study, a small number of patients exhibited CTCAE grade 1 elevations in liver enzymes following breast radiotherapy, despite only incidental liver exposure. Such mild elevations are generally asymptomatic, transient, and clinically insignificant, requiring no intervention [15]. Similar findings have been reported in prior studies of right-sided breast irradiation, where minor increases in AST, ALT, or ALP were observed without progression to RILD or high-grade hepatotoxicity [16]. These results suggest a minimal subclinical hepatic response rather than overt toxicity. However, since this study aimed to explore the potential for radiation-induced liver disease (RILD), it is noteworthy that even low-grade liver enzyme elevations may serve as early indicators of hepatic sensitivity. Therefore, minimizing even grade 1 LEE through optimized dose constraints may help reduce the risk of developing higher-grade liver toxicity or RILD in vulnerable patients.

Currently, the only established treatment for RILD is supportive care, so it is crucial to use thorough evaluation, preventive measures, and careful treatment planning to minimize the occurrence of RILD [17]. There have been several reports on RT hepatotoxicity and consequential RILD in patients with hepatic malignancies, and these have shown that RILD is significantly associated with the dose to normal liver volume. Dawson and colleagues estimated that the liver doses associated with a 5% risk of developing RILD are 90 Gy for uniform irradiation of one-third of the liver, 47 Gy for two-thirds of the liver, and 31 Gy for the entire liver [9]. However, these constraints are unrealistically high for patients with breast cancer and are therefore inappropriate for extrapolation. In previous studies assessing hepatotoxicity after RT in patients with breast cancer, a trend of LEE was observed, but RILD was not observed [13,14,18]. Similarly, in the current study, the median value of the mean dose to liver was 168.9 cGy, and none of the patients had underlying liver dysfunction. Naturally, the incidence of classic RILD, which is roughly equivalent to grade 1 elevation of ALP, was less than 5%, and the occurrence of RILD was minimal.

We conducted a supplementary dosimetric analysis in four representative cases, using the RTOG and ESTRO guidelines (Appendix A) [11,19]. Although three cases showed slightly reduced V5Gy with ESTRO planning, changes in V10Gy and V20Gy were inconsistent, suggesting that there is no superior guideline in terms of reducing liver dose and highlighting the importance of personalized RT planning strategies.

An effective approach to reducing hepatic radiation exposure during right-sided breast cancer radiotherapy is the use of the deep inspiration breath-hold (DIBH) technique. By increasing thoracic volume, DIBH displaces the liver inferiorly and posteriorly, thereby increasing the distance between the liver and the radiation field. This technique has been shown to significantly decrease liver dose compared to free-breathing (FB) methods. In a recent meta-analysis, Li et al. demonstrated that postoperative RT with DIBH resulted in a substantial reduction in both mean liver dose and liver V20Gy (standardized mean difference ≈ −1.15) compared to FB [20]. Considering that even low-dose liver irradiation has been associated with subclinical LEE, the application of DIBH may contribute to mitigating this risk [13]. Furthermore, DIBH may help reduce the likelihood of developing RILD in susceptible patients. Given its feasibility and reproducibility in clinical settings, DIBH represents a practical and effective liver-sparing strategy, particularly for patients at increased risk of hepatic toxicity.

Aside from the radiation dose delivered to the liver, the use of hepatotoxic chemotherapy and the presence of underlying liver dysfunction are critical in evaluating the risk of hepatotoxicity. Online databases such as LiverTox^®^ outline the hepatotoxic potential of various drugs and substances [21]. Among the drugs frequently used in breast cancer treatment, doxorubicin, cyclophosphamide, tamoxifen, and trastuzumab are classified with a likelihood score of B on LiverTox^®^, indicating a high likelihood of liver injury. Fortunately, most of these drugs are self-limiting in terms of hepatotoxicity, with spontaneous recovery reported. Pre-existing liver disease also increases the risk of RILD. Additionally, in patients with hepatitis B virus infection, reactivation of the virus following irradiation is a potentially fatal complication [22]. In patients with such risk factors, it is important to adopt a conservative approach during RT planning and dose optimization. Although this study excluded patients with underlying liver disease, the use of neoadjuvant therapy was found to be associated with a higher incidence of LEE. Although only a small fraction of LEEs progressed to RILD, care should be taken to minimize radiation doses during planning, and regular LFT monitoring should be conducted during post-treatment follow-up.

There were several limitations in this study. First, this was a non-randomized retrospective study and was therefore subject to selection bias. Second, because the study only included patients treated at two institutions, it may not fully reflect the outcomes of breast cancer patients with a wide range of conditions, or the effects of various treatment modalities used in other institutions. Also, in clinical practice, patients with underlying liver disease have the greatest risk of liver-related adverse effects, but these patients were excluded from this study. This was for two reasons: first, to facilitate interpretation of the analysis results, and second, because the proportion of patients with underlying liver disease was small, making it difficult to obtain statistically meaningful outcomes. Finally, there were very few events compared to the number of patients, and the number of variables analyzed. Thus, the characteristics of each patient with LEE or RILD could have influenced the outcome and potentially increase the uncertainty of the statistical analysis. Despite these limitations, to our knowledge, this study is the largest retrospective study to analyze the effects that RT combined with other modalities has on the liver in patients with breast cancer.

## 5. Conclusions

Patients with breast cancer who underwent multi-modality treatments without underlying liver disease showed limited hepatotoxicity. Caution should be taken when treating patients with underlying liver disease, or patients that receive neoadjuvant systemic therapies. Further studies with large patient groups are warranted to verify the hepatotoxicity of RT in patients with breast cancer.

## Figures and Tables

**Figure 1 cancers-17-03252-f001:**
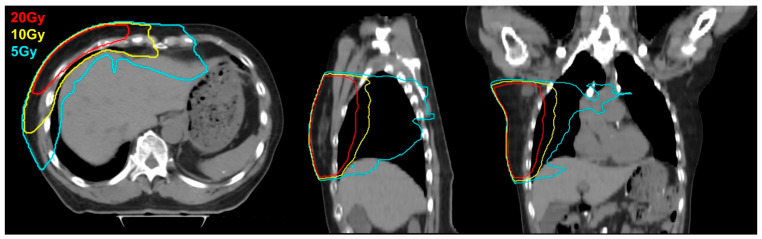
A representative case of breast cancer patient treated with radiotherapy, with 5 Gy, 10 Gy, and 20 Gy isodose line delineated.

**Table 1 cancers-17-03252-t001:** Patient and treatment characteristics.

Characteristics	n = 529	%
**Age, yr, median (range)**	51	(27–86)
**Laterality**		
Right	392	74.1%
Left	137	25.9%
**Pathologic T stage**		
0, in situ	42	7.9%
1	317	59.9%
2	137	25.9%
3	26	4.9%
4	7	1.3%
**Pathologic N stage**		
X	35	6.6%
0	315	59.5%
1	118	22.3%
2	43	8.1%
3	18	3.4%
**Surgery, Primary lesion**		
Breast-conserving surgery	414	78.3%
Mastectomy	115	21.7%
**Surgery, Nodal**		
No axillary surgery	55	10.4%
SLNB	353	66.7%
SLNB with ALND	121	22.9%
**Neoadjuvant therapy**	146	27.6%
**Targeted therapy**	79	14.9%
**Endocrine therapy**	359	67.9%
**Cytotoxic chemotherapy**	261	49.3%
**Adriamycin**	135	25.5%
**Taxane**	257	48.6%
**Platinum-based**	43	8.1%
**Cyclophosphamide**	240	45.4%
**Breast RT dose, Gy, median (range)**	42.56	(38.5–50.06)
**Boost radiotherapy**	321	60.7%
**Boost dose, Gy, median (range)**	9.6	(7.5–17.5)
**Mean dose to liver (cGy)**	168.9	(0.8–984.1)
**V5Gy (%)**	6.1	(0–83.7)
**V10Gy (%)**	0.75	(0–33.0)
**V20Gy (%)**	0.04	(0–12.35)
**IMRT**	414	78.3%
**3D-CRT**	115	21.7%

Abbreviations: 3D-CRT, 3 dimensional-conformal radiotherapy; ALND, Axillary lymph node dissection; IMRT, Intensity modulated radiotherapy; SLNB, Sentinel lymph node biopsy; VnGy, Volume of liver receiving n Gy or greater.

**Table 2 cancers-17-03252-t002:** CTCAE grade of liver enzyme elevation before and after radiotherapy, and the change in CTCAE grade before and after radiotherapy.

	Pre-Radiotherapy		Post-Radiotherapy		*p*		Change *	
	N	*%*	N	*%*			N	*%*
**AST**					<0.001	**AST**		
						−1	4	0.8%
CTCAE grade 0	523	98.9%	521	98.5%		0	519	98.1%
CTCAE grade 1	6	1.1%	8	1.5%		1	6	1.1%
**ALT**					<0.001	**ALT**		
						−1	11	2.1%
CTCAE grade 0	514	97.2%	517	97.7%		0	509	96.2%
CTCAE grade 1	15	2.8%	10	1.9%		1	8	1.5%
CTCAE grade 2	0	0.0%	2	0.4%		2	1	0.2%
**ALP**					<0.001	**ALP**		
						−1	16	3.0%
CTCAE grade 0	506	95.7%	497	94.0%		0	488	92.2%
CTCAE grade 1	23	4.3%	32	6.0%		1	25	4.7%

* Change in CTCAE grade of liver enzyme elevation = CTCAE grade of liver enzyme elevation after radiotherapy-CTCAE grade of liver enzyme elevation before radiotherapy. Abbreviations: ALP, Alkaline phosphatase; ALT, Alanine transferase; AST, Aspartate transferase, CTCAE, Common Terminology Criteria for Adverse Events.

**Table 3 cancers-17-03252-t003:** The median value of DVH parameters with and without LEE. Logistic regression analysis of the association between LEE and DVH parameters.

Characteristics	Elevation		Logistic Regression		
**AST elevation**	**Yes**	**No**	OR	95% CI	*p*
Mean dose (cGy)	151.97	223.49	0.990	0.99–1.00	0.371
V5Gy	7.01	12.98	0.97	0.89–1.04	0.385
V10Gy	1.48	3.86	0.88	0.66–1.17	0.38
V20Gy	0.36	0.86	0.74	0.31–1.79	0.507
**ALT elevation**	**Yes**	**No**	OR	95% CI	*p*
Mean dose (cGy)	223.2	190.4	0.99	0.99–1.00	0.632
V5Gy	12.99	7.79	0.97	0.92–1.03	0.380
V10Gy	3.86	2.47	0.95	0.81–1.11	0.541
V20Gy	0.86	0.59	0.89	0.53–1.50	0.673
**ALP elevation**	**Yes**	**No**	OR	95% CI	*p*
Mean dose (cGy)	293.4	219.2	1	0.99–1.00	0.063
V5Gy	19.15	12.6	1.02	0.99–1.04	0.054
V10Gy	5.72	0.84	1.04	0.98–1.09	0.128
V20Gy	0.99	0.84	1.04	0.85–1.29	0.681
**Any LEE**	**Yes**	**No**	OR	95% CI	*p*
Mean dose (cGy)	263.3	219.8	1	0.99–1.00	0.196
V5Gy	16.2	12.67	1.01	0.99–1.03	0.218
V10Gy	4.83	3.77	1.02	0.98–1.07	0.336
V20Gy	0.85	0.85	1	0.82–1.21	0.996

Abbreviations: ALP, Alkaline phosphatase; ALT, Alanine transferase; AST, Aspartate transferase, CI, Confidence interval; DVH, Dose-volume histogram; LEE, Liver enzyme elevation; OR, Odds ratio; VnGy, Volume of liver receiving n Gy or greater.

**Table 4 cancers-17-03252-t004:** Univariate analysis of the association between LEE of each liver enzyme and DVH parameters.

Characteristics	Logistic Regression		
**AST**	OR	95% CI	*p*
V5Gy (>3.54% vs. <3.54%)	0.76	0.15–3.78	0.732
V10Gy (>1.74% vs. <1.74%)	0.67	0.12–3.68	0.642
V20Gy (>0.01% vs. <0.01%)	3.78	0.44–32.54	0.227
Mean Dose (>135.2cGy vs. <135.2cGy)	0.74	0.15–3.72	0.718
**ALT**	OR	95% CI	*p*
V5Gy (>3.54% vs. <3.54%)	2.30	0.46–11.49	0.311
V10Gy (>1.74% vs. <1.74%)	0.80	0.19–3.39	0.764
V20Gy (>0.01% vs. <0.01%)	2.26	0.45–11.32	0.320
Mean Dose (>135.2cGy vs. <135.2cGy)	2.26	0.45–11.32	0.320
**ALP**	OR	95% CI	*p*
V5Gy (>3.54% vs. <3.54%)	0.76	0.15–3.78	0.732
V10Gy (>1.74% vs. <1.74%)	0.67	0.12–3.68	0.642
V20Gy (>0.01% vs. <0.01%)	3.78	0.44–32.54	0.227
Mean Dose (>135.2cGy vs. <135.2cGy)	0.74	0.15–3.72	0.718
**All**	OR	95% CI	*p*
V5Gy (>3.54% vs. <3.54%)	2.30	1.06–5.01	0.036
V10Gy (>1.74% vs. <1.74%)	1.65	0.83–3.28	0.156
V20Gy (>0.01% vs. <0.01%)	1.68	0.81–3.51	0.166
Mean Dose (>135.2cGy vs. <135.2cGy)	2.26	1.04–4.93	0.040

Abbreviations: ALP, Alkaline phosphatase; ALT, Alanine transferase; AST, Aspartate transferase, CI, Confidence interval; DVH, Dose-volume histogram; LEE, Liver enzyme elevation; OR, Odds ratio; VnGy, Volume of liver receiving n Gy or greater.

**Table 5 cancers-17-03252-t005:** Multivariate analysis of the association between any LEE and DVH parameters. (analyzed by each DVH parameter).

Characteristics	Logistic Regression		
**V5Gy**	**OR**	**95% CI**	** *p* **
**V5Gy (>1.78% vs. <1.78%)**	2.05	0.72–5.83	0.179
**RT Plan (IMRT vs. 3D)**	1.36	0.45–4.16	0.587
**Neoadjuvant therapy (Yes vs. No)**	2.69	1.09–6.65	0.032
**Cyclophosphamide (Yes vs. No)**	0.34	0.06–1.78	0.201
**Taxane (Yes vs. No)**	2.07	0.37–11.49	0.404
**Adriamycin (Yes vs. No)**	2.78	0.84–9.18	0.093
**Platinum (Yes vs. No)**	2.41	0.49–11.92	0.282
**Tamoxifen (Yes vs. No)**	0.42	0.17–1.05	0.065
**Laterality (Right vs. Left)**	2.44	0.82–7.25	0.108
**V10Gy**	**OR**	**95% CI**	** *p* **
**V10Gy (>0.55% vs. <0.55%)**	0.92	0.38–2.26	0.861
**RT Plan (IMRT vs. 3D)**	1.97	0.65–5.95	0.23
**Neoadjuvant therapy (Yes vs. No)**	2.64	1.07–6.50	0.035
**Cyclophosphamide (Yes vs. No)**	0.32	0.06–1.73	0.187
**Taxane (Yes vs. No)**	2.25	0.41–12.45	0.352
**Adriamycin (Yes vs. No)**	3	0.91–9.92	0.072
**Platinum (Yes vs. No)**	2.37	0.47–12.03	0.298
**Tamoxifen (Yes vs. No)**	0.43	0.17–1.06	0.067
**Laterality (Right vs. Left)**	2.91	0.89–9.50	0.077
**V20Gy**	**OR**	**95% CI**	** *p* **
**V20Gy (>0.59% vs. <0.59%)**	0.79	0.35–1.80	0.576
**RT Plan (IMRT vs. 3D)**	1.99	0.71–5.62	0.192
**Neoadjuvant therapy (Yes vs. No)**	2.68	1.09–6.60	0.033
**Cyclophosphamide (Yes vs. No)**	0.31	0.06–1.68	0.175
**Taxane (Yes vs. No)**	2.32	0.42–12.89	0.335
**Adriamycin (Yes vs. No)**	3.04	0.92–10.00	0.067
**Platinum (Yes vs. No)**	2.38	0.47–12.16	0.298
**Tamoxifen (Yes vs. No)**	0.43	0.17–1.06	0.067
**Laterality (Right vs. Left)**	3.09	1.00–9.54	0.050
**Mean Dose**	**OR**	**95% CI**	** *p* **
**Mean Dose** **(>99.6 Gy vs. <99.6 Gy)**	1.95	0.69–5.46	0.206
**RT Plan (IMRT vs. 3D)**	1.35	0.43–4.22	0.604
**Neoadjuvant therapy (Yes vs. No)**	2.72	1.10–6.72	0.030
**Cyclophosphamide (Yes vs. No)**	0.33	0.06–1.71	0.184
**Taxane (Yes vs. No)**	2.15	0.39–11.71	0.377
**Adriamycin (Yes vs. No)**	2.78	0.85–9.16	0.092
**Platinum (Yes vs. No)**	2.4	0.49–11.87	0.282
**Tamoxifen (Yes vs. No)**	0.43	0.17–1.06	0.068
**Laterality (Right vs. Left)**	2.7	0.92–7.95	0.071

Abbreviations: 3D-CRT, 3 dimensional-conformal radiotherapy; ALP, Alkaline phosphatase; ALT, Alanine transferase; AST, Aspartate transferase, CI, Confience interval; DVH, Dose-volume histogram; IMRT, Intensity modulated radiotherapy; LEE, Liver enzyme elevation; OR, Odds ratio; RT, Radiotherapy; VnGy, Volume of liver receiving n Gy or greater.

## Data Availability

The data used in this study are not publicly available as they contain information that could compromise patient privacy. However, they may be made available upon reasonable request.

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
