# Peer review of "Comprehensive Evaluation of Hepatotoxicity Following Radiation Therapy in Breast Cancer Patients"

_cancers, 2025, doi:10.3390/cancers17193252_

Round 1

Reviewer 1 Report

Comments and Suggestions for Authors

This is retrospective study to investigate the impact of liver function after breast radiotherapy.  The authors did a thorough data collection and analysis.  The results showed clinically insignificant impact on liver function after breast radiotherapy.  A few comments and suggestions:

  1. Doses to the liver may differ depending on the side of treatment (right breast vs. left breast) or techniques of treatment (IMRT, breath hold, high tangent, etc.).  Recommend further subgroup analysis to address these.
  2. Any long term impact of LFT abnormality?  Did this finding resolve and if so what is the duration?  Does this impact subsequent systemic therapy treatment?

Overall this is a thorough retrospective review of this topic and is well written.

Author Response

# Reviewer 1

This is retrospective study to investigate the impact of liver function after breast radiotherapy. The authors did a thorough data collection and analysis. The results showed clinically insignificant impact on liver function after breast radiotherapy. A few comments and suggestions:

  1. Doses to the liver may differ depending on the side of treatment (right breast vs. left breast) or techniques of treatment (IMRT, breath hold, high tangent, etc.). Recommend further subgroup analysis to address these.

>> We appreciate your comment. We have added the laterality of treatment (right vs. left breast) as a factor in the multivariate analysis in Table 5 (Shown below). As a result, the laterality of the treatment was not significantly associated with LEE events. Also, the planning technique (IMRT vs. 3D) was already included as a factor in the multivariate analysis. Breath-hold techniques were not used in the treatment of patients, and thus could not be incorporated into the analysis as a factor.

Table 5 Multivariate analysis of the association between any LEE and DVH parameters. (analyzed by each DVH parameter)

Characteristics

Logistic regression

V5Gy

OR

95% CI

P

V5Gy (>1.78% vs. <1.78%)

2.05

0.72 - 5.83

0.179

RT Plan (IMRT vs. 3D)

1.36

0.45 - 4.16

0.587

Neoadjuvant therapy (Yes vs. No)

2.69

1.09 - 6.65

0.032

Cyclophosphamide (Yes vs. No)

0.34

0.06 - 1.78

0.201

Taxane (Yes vs. No)

2.07

0.37 - 11.49

0.404

Adriamycin (Yes vs. No)

2.78

0.84 - 9.18

0.093

Platinum (Yes vs. No)

2.41

0.49 - 11.92

0.282

Tamoxifen (Yes vs. No)

0.42

0.17 - 1.05

0.065

Laterality (Right vs. Left)

2.44

0.82 - 7.25

0.108

V10Gy

OR

95% CI

P

V10Gy (>0.55% vs. <0.55%)

0.92

0.38 - 2.26

0.861

RT Plan (IMRT vs. 3D)

1.97

0.65 - 5.95

0.23

Neoadjuvant therapy (Yes vs. No)

2.64

1.07 - 6.50

0.035

Cyclophosphamide (Yes vs. No)

0.32

0.06 - 1.73

0.187

Taxane (Yes vs. No)

2.25

0.41 - 12.45

0.352

Adriamycin (Yes vs. No)

3

0.91 - 9.92

0.072

Platinum (Yes vs. No)

2.37

0.47 - 12.03

0.298

Tamoxifen (Yes vs. No)

0.43

0.17 - 1.06

0.067

Laterality (Right vs. Left)

2.91

0.89 - 9.50

0.077

V20Gy

OR

95% CI

P

V20Gy (>0.59% vs. <0.59%)

0.79

0.35 - 1.80

0.576

RT Plan (IMRT vs. 3D)

1.99

0.71 - 5.62

0.192

Neoadjuvant therapy (Yes vs. No)

2.68

1.09 - 6.60

0.033

Cyclophosphamide (Yes vs. No)

0.31

0.06 - 1.68

0.175

Taxane (Yes vs. No)

2.32

0.42 - 12.89

0.335

Adriamycin (Yes vs. No)

3.04

0.92 - 10.00

0.067

Platinum (Yes vs. No)

2.38

0.47 - 12.16

0.298

Tamoxifen (Yes vs. No)

0.43

0.17 - 1.06

0.067

Laterality (Right vs. Left)

3.09

1.00 - 9.54

0.050

Mean Dose

OR

95% CI

P

Mean Dose
(>99.6Gy vs. <99.6Gy)

1.95

0.69 - 5.46

0.206

RT Plan (IMRT vs. 3D)

1.35

0.43 - 4.22

0.604

Neoadjuvant therapy (Yes vs. No)

2.72

1.10 - 6.72

0.030

Cyclophosphamide (Yes vs. No)

0.33

0.06 - 1.71

0.184

Taxane (Yes vs. No)

2.15

0.39 - 11.71

0.377

Adriamycin (Yes vs. No)

2.78

0.85 - 9.16

0.092

Platinum (Yes vs. No)

2.4

0.49 - 11.87

0.282

Tamoxifen (Yes vs. No)

0.43

0.17 - 1.06

0.068

Laterality (Right vs. Left)

2.7

0.92 - 7.95

0.071

Abbreviations: 3D-CRT, 3 dimensional-conformal radiotherapy; ALP, Alkaline phosphatase; ALT, Alanine transferase; AST, Aspartate transferase, CI, Confience interval; DVH, Dose-volume histogram; IMRT, Intensity modulated radiotherapy; LEE, Liver enzyme elevation; OR, Odds ratio; RT, Radiotherapy; VnGy, Volume of liver receiving n Gy or greater.

  1. Any long term impact of LFT abnormality? Did this finding resolve and if so what is the duration? Does this impact subsequent systemic therapy treatment?

>> We appreciate your comment. In our currenty study, low grade LFT abnormality rarely caused long term impact. However, patients diagnosed with RILD had lasting effects. Several months of conservative treatment were needed, and systemic therapy were halted in the process. The specific details can be found in this paragraph of the article.:

“Two patients were diagnosed with non-classic RILD. Upon diagnosis, systemic therapy was discontinued in both cases. One patient was treated with ursodeoxycholic acid (UDCA) and silymarin, while the other received UDCA in combination with Godex® capsules. Ultrasonography was performed in one patient ten days after the diagnosis, which revealed diffuse fatty liver. One patient who was diagnosed with classic RILD was admitted for generalized edema with seizure-like movement. But no specific treatment was given for RILD. It resolved concurrently with the improvement of the underlying diseases, and no imaging was done for RILD.”

Overall this is a thorough retrospective review of this topic and is well written.

Reviewer 2 Report

Comments and Suggestions for Authors

First of all, I would like to congratulate the authors for the thorough effort invested in this manuscript. The conclusions are clearly formulated and provide valuable insights for the radiotherapy community, as they highlight strategies aimed at improving patient safety and minimizing treatment-related toxicities.

The methodology is accurately presented; however, I would like to raise a few minor points. In the current context, including patients with left-sided breast cancer may have influenced the results, as liver exposure during left-sided irradiation is generally negligible. The evidence might have been more impactful and precise had the analysis been restricted to right-sided cases. I would therefore suggest that the authors include a justification for this choice in the discussion section.

Additionally, as the authors themselves acknowledged, excluding patients with preexisting liver disease may limit the generalizability of the findings, since real-world clinical outcomes could differ in this subgroup.

Nevertheless, this work remains highly valuable. It offers practical guidance for clinicians, emphasizing the role of deep inspiration breath-hold (DIBH) in reducing liver dose during right-sided breast irradiation, and underscoring the need for closer monitoring of patients receiving neoadjuvant therapy in order to mitigate the risk of liver injury from cumulative toxicities.

Author Response

# Reviewer 2

First of all, I would like to congratulate the authors for the thorough effort invested in this manuscript. The conclusions are clearly formulated and provide valuable insights for the radiotherapy community, as they highlight strategies aimed at improving patient safety and minimizing treatment-related toxicities.

The methodology is accurately presented; however, I would like to raise a few minor points. In the current context, including patients with left-sided breast cancer may have influenced the results, as liver exposure during left-sided irradiation is generally negligible. The evidence might have been more impactful and precise had the analysis been restricted to right-sided cases. I would therefore suggest that the authors include a justification for this choice in the discussion section.

>> We appreciate your thoughtful comment. We agree with your point that the laterality of the tumor, and thus the treatment can influence the outcome of the current study. Thus, we have added laterality as a factor in the multivariate analysis in Table 5. Laterality has shown no statistically significant difference in the incidence of LEE events.:

Table 5 Multivariate analysis of the association between any LEE and DVH parameters. (analyzed by each DVH parameter)

Characteristics

Logistic regression

V5Gy

OR

95% CI

P

V5Gy (>1.78% vs. <1.78%)

2.05

0.72 - 5.83

0.179

RT Plan (IMRT vs. 3D)

1.36

0.45 - 4.16

0.587

Neoadjuvant therapy (Yes vs. No)

2.69

1.09 - 6.65

0.032

Cyclophosphamide (Yes vs. No)

0.34

0.06 - 1.78

0.201

Taxane (Yes vs. No)

2.07

0.37 - 11.49

0.404

Adriamycin (Yes vs. No)

2.78

0.84 - 9.18

0.093

Platinum (Yes vs. No)

2.41

0.49 - 11.92

0.282

Tamoxifen (Yes vs. No)

0.42

0.17 - 1.05

0.065

Laterality (Right vs. Left)

2.44

0.82 - 7.25

0.108

V10Gy

OR

95% CI

P

V10Gy (>0.55% vs. <0.55%)

0.92

0.38 - 2.26

0.861

RT Plan (IMRT vs. 3D)

1.97

0.65 - 5.95

0.23

Neoadjuvant therapy (Yes vs. No)

2.64

1.07 - 6.50

0.035

Cyclophosphamide (Yes vs. No)

0.32

0.06 - 1.73

0.187

Taxane (Yes vs. No)

2.25

0.41 - 12.45

0.352

Adriamycin (Yes vs. No)

3

0.91 - 9.92

0.072

Platinum (Yes vs. No)

2.37

0.47 - 12.03

0.298

Tamoxifen (Yes vs. No)

0.43

0.17 - 1.06

0.067

Laterality (Right vs. Left)

2.91

0.89 - 9.50

0.077

V20Gy

OR

95% CI

P

V20Gy (>0.59% vs. <0.59%)

0.79

0.35 - 1.80

0.576

RT Plan (IMRT vs. 3D)

1.99

0.71 - 5.62

0.192

Neoadjuvant therapy (Yes vs. No)

2.68

1.09 - 6.60

0.033

Cyclophosphamide (Yes vs. No)

0.31

0.06 - 1.68

0.175

Taxane (Yes vs. No)

2.32

0.42 - 12.89

0.335

Adriamycin (Yes vs. No)

3.04

0.92 - 10.00

0.067

Platinum (Yes vs. No)

2.38

0.47 - 12.16

0.298

Tamoxifen (Yes vs. No)

0.43

0.17 - 1.06

0.067

Laterality (Right vs. Left)

3.09

1.00 - 9.54

0.050

Mean Dose

OR

95% CI

P

Mean Dose
(>99.6Gy vs. <99.6Gy)

1.95

0.69 - 5.46

0.206

RT Plan (IMRT vs. 3D)

1.35

0.43 - 4.22

0.604

Neoadjuvant therapy (Yes vs. No)

2.72

1.10 - 6.72

0.030

Cyclophosphamide (Yes vs. No)

0.33

0.06 - 1.71

0.184

Taxane (Yes vs. No)

2.15

0.39 - 11.71

0.377

Adriamycin (Yes vs. No)

2.78

0.85 - 9.16

0.092

Platinum (Yes vs. No)

2.4

0.49 - 11.87

0.282

Tamoxifen (Yes vs. No)

0.43

0.17 - 1.06

0.068

Laterality (Right vs. Left)

2.7

0.92 - 7.95

0.071

Abbreviations: 3D-CRT, 3 dimensional-conformal radiotherapy; ALP, Alkaline phosphatase; ALT, Alanine transferase; AST, Aspartate transferase, CI, Confience interval; DVH, Dose-volume histogram; IMRT, Intensity modulated radiotherapy; LEE, Liver enzyme elevation; OR, Odds ratio; RT, Radiotherapy; VnGy, Volume of liver receiving n Gy or greater.

Additionally, as the authors themselves acknowledged, excluding patients with preexisting liver disease may limit the generalizability of the findings, since real-world clinical outcomes could differ in this subgroup.

>> We appreciate your comment. The exclusion of patients with underlying hepatic disease may constrain the external validity of the study. Nonetheless, this decision was made primarily to ensure the clear interpretation of the analysis results, and secondarily, because the limited number of patients with pre-existing liver disease precluded obtaining statistically robust outcomes. We have mentioned this in the discussion section as follows:

“Also, in clinical practice, patients with underlying liver disease have the greatest risk of liver-related adverse effects, but these patients were excluded from this study. This was for two reasons: first, to facilitate interpretation of the analysis results, and second, because the proportion of patients with underlying liver disease was small, making it difficult to obtain statistically meaningful outcomes.”

Nevertheless, this work remains highly valuable. It offers practical guidance for clinicians, emphasizing the role of deep inspiration breath-hold (DIBH) in reducing liver dose during right-sided breast irradiation, and underscoring the need for closer monitoring of patients receiving neoadjuvant therapy in order to mitigate the risk of liver injury from cumulative toxicities.
